# Exploring the Intersection of Microplastics and Black Soldier Fly Larvae: A Comprehensive Review

**DOI:** 10.3390/insects16090913

**Published:** 2025-09-01

**Authors:** Claudiu-Nicusor Ionica, Romelia Pop, Raluca Paula Popa, Alexandru-Flaviu Tabaran, Dragos Hodor, Sergiu Condor, Sorana Daina, Andrei-Radu Szakacs, Adrian Macri

**Affiliations:** 1Department of Animal Nutrition, Faculty of Veterinary Medicine, University of Agricultural Sciences and Veterinary Medicine of Cluj-Napoca, Calea Manaștur, 400372 Cluj-Napoca, Romania; claudiu-nicusor.ionica@usamvcluj.ro (C.-N.I.); andrei.szakacs@usamvcluj.ro (A.-R.S.); adrian.macri@usamvcluj.ro (A.M.); 2Department of Pathology, Faculty of Veterinary Medicine, University of Agricultural Sciences and Veterinary Medicine of Cluj-Napoca, Calea Manaștur, 400372 Cluj-Napoca, Romania; raluca-paula.popa@student.usamvcluj.ro (R.P.P.); alexandru.tabaran@usamvcluj.ro (A.-F.T.); dragos.hodor@usamvcluj.ro (D.H.); 3Department of Animal Breeding and Food Science, Faculty of Veterinary Medicine, University of Agricultural Sciences and Veterinary Medicine, Manastur Street No. 3/5, 400372 Cluj-Napoca, Romania; sergiu.condor@student.usamvcluj.ro

**Keywords:** microplastic, *Hermetia illucens*, bioaccumulation, environmental impact, toxic, particles

## Abstract

This systematic review explores the potential of Black Soldier Fly Larvae (BSFL) as a tool for bioremediation of microplastic pollution. The core of this study focuses on the larvae’s ability to reduce microplastics through bioaccumulation and degradation. The study highlights that while BSFL show promise in this area, the specific mechanisms are not yet fully understood. It concludes that further research is essential to optimize these bioremediation strategies and assess their long-term environmental impact, particularly given the inconsistencies found in current experimental data.

## 1. Introduction

The persistence of microplastic pollution in ecosystems has become a major environmental concern, with microplastics being identified in terrestrial, freshwater, and marine environments at concerning rates [1]. Every year, global plastic production rises, reaching 320 million tons annually [2]. It was estimated that more than 10 million metric tons of plastic waste are released into the oceans annually [3]. Although some plastics enter the oceans through maritime activities, it is believed that 80% of them come from land-based sources [4]. These pollutants originate from the degradation of larger plastic debris and the release of primary microplastics, contributing to a wide range of adverse ecological and biological impacts, including their potential to act as vectors for harmful substances like heavy metals and organic pollutants (polychlorinated biphenyls, polycyclic aromatic hydrocarbons, and organochlorine pesticides like dichlorodiphynyltrichloroethane) [5,6]. Despite growing awareness, effective methods for reducing the spread of microplastics are still in the developmental phase. In recent years, the use of biological systems for waste remediation has gained traction as an environmentally friendly alternative to traditional methods [7,8]. Among these, the Black Soldier Fly Larvae (BSFL) have been highlighted as a promising agent for organic waste bioremediation due to their ability to break down organic materials, including complex compounds like pharmaceuticals and hydrocarbons [9,10]. BSFL offer a dual function in waste management: they can bioaccumulate contaminants, aiding in the safe disposal of hazardous waste, while also being a sustainable resource for industries through nutrient recovery and waste valorization [11,12]. The potential of BSFL to interact with and degrade microplastic particles has gathered increasing attention. While BSFL has demonstrated efficacy in bioaccumulating microplastics, their mechanisms of degradation and the impact of these interactions on the larvae and the surrounding microbiome remain underexplored [13]. Studies suggest that microbial communities associated with different insect larvae may play a critical role in facilitating microplastic degradation, thus adding complexity to the bioremediation process [14,15].Moreover, alterations in the BSFL gut microbiome due to microplastic ingestion have been shown to influence the efficiency of organic waste breakdown, which could provide new insights into optimizing bioremediation strategies [16]. However, research in this area is still at an early stage, and numerous challenges remain. Current studies face limitations such as inconsistencies in experimental design, variations in microplastic types and concentrations, and differing environmental conditions, all of which hinder the comparison of findings [1]. Additionally, there is a need to better understand the long-term environmental and ecological consequences of using BSFL in microplastic remediation, including the fate of microplastics post-bioremediation [17,18]. This systematic review aims to critically evaluate the existing body of research on the interaction between BSFL and microplastic particles, with a focus on understanding the digestion and degradation mechanisms, microbiome changes, and the challenges currently faced in this field of study. By synthesizing the findings from recent studies, we seek to provide a comprehensive overview of the role of BSFL in microplastic bioremediation and identify key areas for future research.

## 2. Methods

### 2.1. Literature Search

The literature search was conducted by the Preferred Reporting Items for Systematic Reviews and Meta-Analyses (PRISMA) guidelines, 2024. We searched the databases BASE (www.base-search.net, last accessed on 5 September 2024), CORE (core.ac.uk, last accessed on 5 September 2024), Google Scholar (scholar.google.com, last accessed on 5 September 2024), ScienceDirect (www.sciencedirect.com, last accessed on 5 September 2024), and Semantic Scholar (semanticscholar.org, last accessed on 5 September 2024). The data were systematically searched in the following databases: BASE (“All databases” selected), CORE (“All databases” selected), Google Scholar (last accessed on 5 September 2024), and Science Direct (“All databases” selected), and Semantic Scholar (“All databases” selected) using keywords “Black soldier fly larvae AND styrene AND toxicity AND polystyrene microplastics”, “BSFL AND styrene AND toxicity AND polystyrene microplastics”, “BSF AND styrene AND toxicity AND polystyrene microplastics”, “*Hermetia illucens* AND styrene AND toxicity AND polystyrene microplastics”, “*H. illucens* AND styrene AND toxicity AND polystyrene microplastics”, “Black soldier fly larvae AND toxicity AND polyamine microplastics”, “BSFL AND toxicity AND polyamine microplastics”, “BSF AND toxicity AND polyamine microplastics”, “*Hermetia illucens* AND toxicity AND polyamine microplastics”, “*H. illucens* AND toxicity AND polyamine microplastics”, “Black soldier fly larvae AND toxicity AND polylactic acid microplastics”, “BSFL AND toxicity AND polylactic acid microplastics”, “BSF AND toxicity AND polylactic acid microplastics”, “*Hermetia illucens* AND toxicity AND polylactic acid microplastics”, “*H. illucens* AND toxicity AND polylactic acid microplastics”, “Black soldier fly larvae AND toxicity AND polyethylene microplastics”, “BSFL AND toxicity AND polyethylene microplastics”, “BSF AND toxicity AND polyethylene microplastics”, “*Hermetia illucens* AND toxicity AND polyethylene microplastics”, “*H. illucens* AND toxicity AND polyethylene microplastics”. The data were searched separately by two of the researchers (I.C.N. and R.P.). Through initial procedure, 1127 items were found: BASE—6 results (“Black soldier fly larvae AND styrene AND toxicity AND polystyrene microplastics”, n = 1; “*Hermetia illucens* AND styrene AND toxicity AND polystyrene microplastics”, n = 1; “H. illucens AND styrene AND toxicity AND polystyrene microplastics”, n = 1; “*H. illucens* AND toxicity AND polyamine microplastics”, n = 1; “Black soldier fly larvae AND toxicity AND polyethylene microplastics”, n = 1; “ *H. illucens* AND toxicity AND polyethylene microplastics”, n = 1); CORE—209 results (“Black soldier fly larvae AND styrene AND toxicity AND polystyrene microplastics”, n = 7; “*Hermetia illucens* AND styrene AND toxicity AND polystyrene microplastics”, n = 5; “*H. illucens* AND styrene AND toxicity AND polystyrene microplastics”, n = 5; “Black soldier fly larvae AND toxicity AND polyamine microplastics”, n = 4; “BSF AND toxicity AND polyamine microplastics”, n = 6; “ *Hermetia illucens* AND toxicity AND polyamine microplastics”, n = 24, “*H. illucens* AND toxicity AND polyamine microplastics, n = 25, “Black soldier fly larvae AND toxicity AND polylactic acid microplastics”, n = 7, “BSFL AND toxicity AND polylactic acid microplastics”, n = 3, “BSF AND toxicity AND polylactic acid microplastics”, n = 2, “*Hermetia illucens* AND toxicity AND polylactic acid microplastics”, n = 5, “Black soldier fly larvae AND toxicity AND polyethylene microplastics”, n = 37, “BSFL AND toxicity AND polyethylene microplastics”, n = 6, “BSF AND toxicity AND polyethylene microplastics”, n = 14, “*Hermetia illucens* AND toxicity AND polyethylene microplastics”, n = 29, “*H. illucens* AND toxicity AND polyethylene microplastics”, n = 30,); Google Scholar (only title were searched)—423 results were found (“Black soldier fly larvae AND styrene AND toxicity AND polystyrene microplastics”, n = 72; “BSFL AND styrene AND toxicity AND polystyrene microplastics”, n = 3; “BSF AND styrene AND toxicity AND polystyrene microplastics”, n = 10; “*Hermetia illucens* AND styrene AND toxicity AND polystyrene microplastics”, n = 32; “*H. illucens* AND styrene AND toxicity AND polystyrene microplastics”, n = 29, “Black soldier fly larvae AND toxicity AND polyamine microplastics”, n=7, “BSF AND toxicity AND polyamine microplastics”, n = 7, “*H. illucens* AND toxicity AND polyamine microplastics”, n = 27 “Black soldier fly larvae AND toxicity AND polylactic acid microplastics”, n = 71, “BSFL AND toxicity AND polylactic acid microplastics”, n = 9, “BSF AND toxicity AND polylactic acid microplastics”, n = 35, “*Hermetia illucens* AND toxicity AND polylactic acid microplastics”, n = 58, “*H. illucens* AND toxicity AND polylactic acid microplastics”, n = 6, “BSFL AND toxicity AND polyethylene microplastics”, n = 57,); Science Direct- 163 (“Black soldier fly larvae AND styrene AND toxicity AND polystyrene microplastics”, n = 7, “BSFL AND styrene AND toxicity AND polystyrene microplastics”, n = 1, “BSF AND styrene AND toxicity AND polystyrene microplastics”, n = 2, “*Hermetia illucens* AND styrene AND toxicity AND polystyrene microplastics”, n = 7, “*H. illucens* AND styrene AND toxicity AND polystyrene microplastics”, n = 6, “BSF AND toxicity AND polyamine microplastics”, n = 2, “*Hermetia illucens* AND toxicity AND polyamine microplastics”, n = 21, “Black soldier fly larvae AND toxicity AND polylactic acid microplastics”, n = 7, “BSFL AND toxicity AND polylactic acid microplastics”, n = 1, “BSF AND toxicity AND polylactic acid microplastics”, n = 6, “*Hermetia illucens* AND toxicity AND polylactic acid microplastics”, n = 6, “Black soldier fly larvae AND toxicity AND polyethylene microplastics”, n = 29, “BSFL AND toxicity AND polyethylene microplastics”, n = 4, “BSF AND toxicity AND polyethylene microplastics”, n = 14, “*Hermetia illucens* AND toxicity AND polyethylene microplastics”, n = 26, “*H. illucens* AND toxicity AND polyethylene microplastics”, n = 24); Semantic Scholar–326 results (“Black soldier fly larvae AND styrene AND toxicity AND polystyrene microplastics”, n = 2, “BSFL AND styrene AND toxicity AND polystyrene microplastics”, n = 7, “BSF AND styrene AND toxicity AND polystyrene microplastics”,n = 7, “*Hermetia illucens* AND styrene AND toxicity AND polystyrene microplastics”, n = 9, “ *H. illucens* AND styrene AND toxicity AND polystyrene microplastics”, n = 80, “Black soldier fly larvae AND toxicity AND polyamine microplastics”, n = 30, “BSFL AND toxicity AND polyamine microplastics”, n = 1, “*Hermetia illucens* AND toxicity AND polyamine microplastics”, n = 2, “*H. illucens* AND toxicity AND polyamine microplastics”, n = 60, “Black soldier fly larvae AND toxicity AND polylactic acid microplastics”,n = 6, “BSFL AND toxicity AND polylactic acid microplastics”, n = 20, “BSF AND toxicity AND polylactic acid microplastics”, n = 20, “*Hermetia illucens* AND toxicity AND polylactic acid microplastics”, n = 30, “*H. illucens* AND toxicity AND polylactic acid microplastics”, n = 20, “Black soldier fly larvae AND toxicity AND polyethylene microplastics”, n = 30, “Hermetia illucens AND toxicity AND polyethylene microplastics”, n = 2,) (Appendix A).

### 2.2. Selection of Studies

The detailed search strategy and corresponding results are presented in the PRISMA diagram (Appendix A). A total of 669 duplicate records were removed. An additional 450 records were excluded for the following reasons: absence of references to “Black soldier fly larvae” or “*Hermetia illucens*” (n = 183); no mention of “microplastics” or specific microplastic types (n = 222); and absence of both “Black soldier fly larvae” and “microplastics” (n = 45). The remaining 8 studies met the inclusion criteria and were subsequently subjected to data extraction and analysis.

## 3. Microplastics and Nanoplastics: Sources, Types, and Environmental Impact

Microplastics represent plastic particles typically less than 5 mm in size [19], originating from various sources that become pervasive in the environment. They can be broadly categorized into primary and secondary microplastics [20,21]. Primary microplastics are manufactured at microscopic sizes for specific applications, such as microbeads in personal care products, industrial abrasives, and fibers shed from synthetic textiles during washing [22]. These are directly released into the environment through household drains or industrial effluents. Secondary microplastics, on the other hand, result from the breakdown of larger plastic debris, such as bottles, bags, and fishing nets, due to physical, chemical, and biological processes like UV radiation, wave action, and microbial degradation [23]. These degradation processes fragment the larger plastics into increasingly smaller pieces, contributing significantly to the accumulation of microplastics in terrestrial [24] and aquatic ecosystems [25]. Additionally, tire wear particles from vehicles, road markings [26], and paint are significant sources of microplastics, especially in urban environments [27]. Together, these sources contribute to the widespread presence of microplastics in the environment, posing serious ecological and health risks due to their persistence, bioaccumulation potential, and ability to adsorb harmful pollutants [28].

Microplastics are categorized based on their size, shape, polymer type, and origin, each of which plays a crucial role in determining their environmental behavior, interaction with organisms, and potential risks. Plastics remain in the environment for extended periods and can break down into smaller fragments, creating various types of plastic debris. These range from macroplastics (larger than 2 cm) and mesoplastics (5 mm to 2 cm), to microplastics (less than 5 mm), and even nanoplastics (smaller than 1 μm) [29]. Microplastics and nanoplastics exhibit various shapes, including fibers, fragments, films, foams [30,31], and beads [32]. Fibers are thread-like microplastics commonly shed from synthetic textiles during washing [33,34]. Fragments are irregularly shaped pieces resulting from the breakdown of larger plastics [35]. Films often come from degraded plastic bags or packaging materials [36], while foams typically originate from polystyrene products like insulation or packaging [37]. Beads are spherical particles, often found in personal care products and as industrial abrasives [38]. The polymer composition of microplastics varies widely, influencing their physical and chemical properties [39]. Common polymers include polyethylene (PE) ((C_2_H_4_)_n_), polypropylene (PP) ((C_3_H_6_)_n_), polystyrene (PS) ((C_8_H_8_)_n_), polyvinyl chloride (PVC) (C_2_H_3_Cl), and polyethylene terephthalate (PET) ((C_10_H_8_O_4_)_n_) [40]. Each polymer type has distinct characteristics, such as density, hydrophobicity, and resistance to degradation, which affect their environmental persistence and interaction with pollutants [41]. For example, polyethylene, commonly used in plastic bags and bottles, is lightweight and tends to float in aquatic environments, making it highly mobile and widespread [42]. From 1990 to 2019, (Figure 1) the global production of plastics amounted to 72.81 million tonnes of polypropylene (PP), 21.12 million tonnes of polystyrene (PS), 51.39 million tonnes of polyvinyl chloride (PVC), and 24.92 million tonnes of polyethylene terephthalate (PET) [43].

### 3.1. Environmental Distribution and Bioaccumulation

Microplastics have become ubiquitous in the environment, accumulating across various ecosystems due to their persistence and the extensive use of plastics in modern society. These particles are transported through different environmental media, including air, water, and soil, leading to their widespread distribution. In aquatic environments, microplastics are found from the surface waters to the deep sea, accumulating in ocean gyres, coastal sediments, and freshwater bodies [1]. Their distribution is influenced by factors such as water currents, wind patterns, and the density of the particles, with lighter particles often remaining suspended in water columns and heavier particles settling into sediments [44].

In terrestrial environments, microplastics accumulate in soils, particularly in agricultural fields where sewage sludge is introduced as fertilizer, which often contains plastic contaminants. Studies showed that microplastics can pass into plant cells via endocytosis [45]. Additionally, urban runoff and atmospheric deposition contribute to soil contamination, leading to the infiltration of microplastics into groundwater systems. The atmosphere also plays a crucial role in the transport of microplastics [46], with particles being carried over long distances by wind, leading to their deposition even in remote areas such as the Arctic and high-altitude mountains [47].

The accumulation of microplastics in these environments is a growing concern due to their potential to act as vectors for chemical pollutants [48] and their long-term persistence [49]. In aquatic systems, microplastics can be ingested by a wide range of organisms, from zooplankton to fish [50], leading to their integration into the food web and potential biomagnification [51]. Microplastics can affect soil health on land by altering soil structure, affecting microbial communities, and potentially entering the food chain through soil-dwelling organisms [52]. The internalization and uptake of nano- and microplastics in the body occur through a range of cellular mechanisms, often influenced by particle size. Primary uptake pathways include enterocyte-mediated endocytosis and transcytosis via M-cells, which are found in Peyer’s patches of the gut-associated lymphoid tissue (GALT). Persorption, involving the movement of particles through gaps left by lost enterocytes at villi tips, and paracellular uptake, where particles move between cells, also facilitate particle absorption. Within cells, clathrin-mediated endocytosis is a major pathway, especially for nanoparticles, while caveolin-mediated endocytosis is prominent in endothelial cells. These pathways involve both energy-dependent processes like phagocytosis and pinocytosis—wherein phagocytosis is typically carried out by immune cells such as macrophages—and non-energy-dependent mechanisms. Additional routes, such as macropinocytosis and dynamin-independent endocytosis, contribute to nanoparticle uptake, involving the Rho GTPases and actin cytoskeleton to form specific vesicles within the cell, facilitating particle transport and localization [53,54,55].

### 3.2. Characteristics Influencing Environmental Impact

The environmental behavior and interactions of microplastics with living organisms are greatly influenced by their characteristics, including size, surface area, and chemical makeup. These factors determine how microplastics behave in various ecosystems and how they may affect biological systems. [56]. Smaller particles, particularly those in the nano-size range, can penetrate biological membranes, potentially leading to cellular toxicity [57]. Nanoplastics can cross biological barriers and have been detected in the human placenta, raising concerns about potential impacts on fetal development and maternal health. Studies suggest that these particles may induce oxidative stress and inflammation, potentially interfering with cellular functions critical for a healthy pregnancy [58,59,60]. The large surface area-to-volume ratio of microplastics allows them to adsorb a variety of persistent organic pollutants (POPs) [61] and heavy metals [62] from the environment, effectively acting as carriers for these harmful substances [63]. The hydrophobic nature of most plastics also enables them to concentrate lipophilic contaminants [64], which can bioaccumulate and magnify up the food chain, posing a significant threat to aquatic organisms and, ultimately, to human health [65].

Also, the durability of microplastics, stemming from their resistance to natural degradation processes, is a major environmental concern [66]. Most microplastics are highly resistant to photodegradation [67], chemical breakdown [68], and microbial attack [69], allowing them to persist in the environment for decades or even centuries [70]. This persistence, combined with their small size and widespread distribution, makes them particularly difficult to remove from natural ecosystems, leading to their accumulation in various habitats, including oceans [71], soils [72], and even the atmosphere [73].

The types and characteristics of microplastics are diverse, influencing their environmental fate, interactions with pollutants and organisms, and potential risks to ecosystems and human health [8]. Understanding these aspects is crucial for developing effective strategies to reduce their impact and manage plastic pollution [74].

### 3.3. Toxic Impacts of Microplastics on Living Organisms

Microplastics represent a significant environmental hazard due to their universal presence across diverse ecosystems. Aquatic organisms are particularly susceptible to microplastic contamination [75]. Marine and freshwater species, including fish, mollusks, and crustaceans, frequently ingest microplastics, mistaking them for organic matter [76]. Once internalized, these particles can obstruct digestive tracts, impair nutrient assimilation, and induce oxidative stress [3]. Terrestrial ecosystems are equally affected. Microplastic contamination in soils alters physicochemical properties [77], disrupting microbial communities [78] essential for nutrient cycling and plant growth [79]. Soil invertebrates, including earthworms [80] and insects, accidentally ingest microplastics, leading to physiological stress, reproductive toxicity, and bioaccumulation [81]. Recent studies indicate that exposure to microplastics in insects, such as pollinators [82] compromises metabolic homeostasis and modifies immune responses [81], ultimately influencing ecological balance and agricultural productivity [83]. Humans are exposed to microplastics via ingestion, inhalation, and dermal contact [84]. Empirical evidence has demonstrated microplastic presence in human circulatory and placental tissues [85], raising concerns regarding inflammatory responses, oxidative stress, endocrine disruption [86], and potential carcinogenic effects [87] mediated by plastic-associated chemical additives, including phthalates and bisphenol A (BPA) [88]. Reducing microplastic pollution requires a multidisciplinary approach [89], including reduced plastic production, biodegradable material development, and more severe regulations [90]. Scientific innovation and public awareness [91] play key roles in minimizing its impact on biodiversity and ecosystem stability.

## 4. Black Soldier Fly Larvae: Biology and Applications

### 4.1. BSFL as a Sustainable Waste Management Solution and Bioremediation

The black soldier fly larvae present an innovative and sustainable approach to organic waste management [92], effectively addressing critical environmental challenges [93]. These larvae exhibit remarkable efficiency in decomposing diverse organic substrates [94], including food waste [95] and agricultural residues [96]. Through their biological processes, BSFL convert organic waste into nutrient-rich biomass [97], which serves as a valuable resource for animal feed [98], biodiesel production [99], and other industrial applications. This waste-to-resource conversion significantly reduces waste accumulation [100] and fosters a circular economy [101] by transforming waste streams into economically beneficial outputs [102]. Additionally, BSFL-based waste processing systems exhibit a low environmental footprint [103], requiring minimal energy and generating fewer greenhouse gas emissions [104] compared to conventional composting methods [105,106]. Due to their adaptability, BSFL can process a broad spectrum of waste materials [107], making them suitable for applications ranging from small-scale agricultural settings [108] to large-scale industrial waste management operations.

Beyond organic waste decomposition, BSFL play a pivotal role in environmental bioremediation, utilizing their metabolic capabilities to reduce environmental pollutants [109]. Their ability to degrade complex organic compounds and bioaccumulate hazardous substances makes them highly suitable for diverse bioremediation applications [110]. Specifically, BSFL have demonstrated effectiveness in processing organic waste containing environmental contaminants such as heavy metals [111], hydrocarbons [112], and pharmaceutical residues [113]. During their feeding process, BSFL can bioaccumulate and sequester toxic metals like cadmium and lead from contaminated waste, thereby reducing their environmental impact [114]. While BSFL may not fully detoxify such contaminants, their ability to concentrate hazardous substances facilitates safer waste disposal and management [115]. Additionally, BSFL have exhibited the potential to degrade persistent organic pollutants, including pesticides and industrial chemicals [116], making them suitable for treating agricultural and industrial waste streams [117].

A particularly promising yet underexplored area is the application of BSFL in plastic bioremediation. Plastics pose a significant global environmental threat due to their persistence and accumulation in ecosystems. Recent studies suggest that certain insect larvae, including BSFL, may harbour gut microbiota capable of facilitating plastic degradation through enzymatic processes [16]. While research on BSFL-mediated plastic biodegradation remains in its early stages, preliminary findings indicate that BSFL can ingest and mechanically fragment plastics, potentially enhancing microbial decomposition [13]. Furthermore, the co-metabolism of plastics with organic waste may improve degradation efficiency by providing essential nutrients for microbial consortia involved in polymer breakdown [118]. The ability of BSFL to process plastic-contaminated waste could be instrumental in reducing plastic pollution. Future research should focus on identifying specific microbial consortia within the BSFL gut that contribute to plastic degradation and optimizing conditions for enhanced biodegradation efficacy.

### 4.2. Life Cycle and Biology of BSFL

The black soldier fly (*Hermetia illucens*) is an insect species native to the Americas but is now found in various regions worldwide due to its adaptability and usefulness in waste management [119]. Its relatively short holometabolous life cycle (approx. 40–45 days, depending on the substrate) [11,120] includes four stages: egg, larva, pupa, and adult, each characterized by specific behaviors and physiological adaptations that contribute to its ecological success [121]. In its natural habitat, the BSF begins its life cycle when the female lays eggs in clusters near decaying organic matter, such as compost or carcasses, providing a nutrient-rich food source for the larvae [122]. The tiny eggs (~1 mm) are laid in hidden cracks or on surfaces near the decaying material, reducing predation and desiccation risks [123]. Egg-laying sites are selected based on strong odours that attract females [124]. The larval stage, lasting 8–10 days, is crucial in the black soldier fly’s life cycle [125]. Larvae inhabit moist, decaying organic matter, acting as efficient decomposers of plant debris and animal remains [126,127]. Their strong mouthparts [128] and digestive enzymes [129] enable the breakdown of tough fibers [130], aiding nutrient recycling [131]. They undergo five to six instars [132] and thrive in microbe-rich environments [120], with gut microbiota potentially enhancing organic decomposition [133]. After the larval stage, black soldier fly larvae enter the pupal stage, undergoing complete metamorphosis [134]. They typically pupate in dry, sheltered locations like leaf litter, soil, or decaying logs, providing protection from predators and environmental stresses [135]. The pupae are encased in a hard exoskeleton for further protection. The pupal stage lasts 1–2 weeks, depending on temperature and humidity [136,137]. The adult black soldier fly has a black body, smoky wings, and white markings on the legs and face (Figure 2). Focused on reproduction, adults are non-feeding and rely on energy reserves from the larval stage [138]. They are commonly found near light sources or at emergence sites, engaging in mating behaviors [139]. Black soldier flies play a crucial role in controlling pest fly populations by consuming decaying organic matter, reducing breeding sites for houseflies [140] and blowflies [141], and helping prevent pest problems in ecosystems and human environments [142].

## 5. Interaction Between Microplastics and Black Soldier Fly Larvae

### 5.1. Ingestion of Microplastics by BSFL

The ingestion rate of microplastics by BSFL is crucial for their use in waste management and bioremediation, as it affects their ability to process or degrade plastics. This rate varies based on factors like the size of the larvae’s mouth apparatus, which determines their efficiency in ingesting microplastic particles. A key finding from Lievens et al. [13] in 2023 demonstrated that the larvae are only able to ingest plastic particles smaller than their mouth opening, which is approximately 110 μm (Figure 3). This factor will influence ingestion, especially the larva’s ability to physically fragment microplastic particles. In experiments where the majority of the plastic particles exceeded this size threshold, the BSFL was unable to ingest them, confirming that the size of the particles plays a significant role in their ingestion. As the larvae grow, the size of their mouth opening increases, starting from about 20 μm at 5 days after hatching (DAH) to around 110 μm at 17 DAH. Lievens et al. [13] observed that by 10 DAH, the larvae’s mouth opening of 65 μm was large enough to ingest microplastics with a median size of 61.5 μm. Despite their ability to grind organic matter, BSFL cannot break down smooth polyethylene microplastics, limiting ingestion to particles smaller than their mouth opening. Microplastic intake depends on particle size, substrate, and larval age. Their mandibular-maxillary apparatus, with a three-toothed hook, grinds food and pushes away larger particles, restricting the ingestion of larger microplastics [128,143]. Further research, like Piersanti’s study in 2024, shows that BSFL interacts with different types of microplastics in varying ways [118]

Lievens et al. [13] found that the larvae had difficulty breaking down smooth polyethylene microplastics, but Piersanti et al. [118] observed that they could ingest PVC microplastics (PVC-MPs). During digestion in the larvae’s midgut, the shape and structure of PVC-MPs were altered, suggesting that BSFL can more easily process irregularly shaped plastics compared to smoother ones. In terms of plasticizers, Lievens et al. [13] reported that BSFL moderately consumed the plasticizer DINP during a 10-day period, with intake ranging from 82 to 273 ng/g. For another plasticizer, DEHT, intake was between 67 and 137 ng/g. Lievens et al. [13] also explained that plasticizers are absorbed by the larvae through migration from plastic materials into the surrounding substrate, which the larvae eat. If plastic particles are too large, like PVC microplastics, larvae cannot ingest them, so plasticizers like DINP are absorbed from the substrate instead. These factors collectively underscore the importance of particle size and type in assessing the potential of BSFL for use in waste management and bioremediation.

### 5.2. Digestion, Degradation, and Microbiome Dynamics in Black Soldier Fly Larvae Exposed to Microplastics

#### 5.2.1. Digestion and Degradation of Microplastics by BSFL

Studies have shown that BSF larvae are capable of ingesting microplastics, such as polyvinyl chloride microplastics (PVC-MPs), during their rearing phase. In a study conducted by Lievens et al. [13] in 2023, BSF larvae were exposed to substrates containing two common plasticizers, DINP and DEHT, in the presence of PVC-MPs [13]. The larvae exhibited a moderate intake of DINP (82–273 ng/g) over 10 days, with biotransformation occurring within 24 h, producing the primary metabolite monoisononyl phthalate (MINP). However, for DEHT, an uptake between 67 and 137 ng/g was observed, but no clear biotransformation pattern emerged. It is important to distinguish between the different processes observed when BSFL interact with microplastics and plasticizers. The alterations in PVC microplastics described in this section should primarily be considered mechanical fragmentation, resulting from abrasion or grinding in the larval gut, which changes particle size and surface morphology but does not alter the polymer backbone. In contrast, biotransformation refers to metabolic modifications of associated plastic additives or low-molecular-weight compounds, such as the conversion of DINP into MINP, which has been documented in BSFL.

The larvae’s digestive tract did not appear to retain substantial amounts of microplastics after excretion, and bioaccumulation factors (BAF) for both DINP and DEHT remained below 1 (BAF < 0.001). This suggests that the larvae were capable of metabolizing or eliminating these compounds shortly after ingestion, preventing significant bioaccumulation. Some microplastic particles, however, were retained in the larvae’s gut even after a period of starvation, highlighting the need for further optimization of starvation protocols to remove residual microplastics before harvesting larvae for potential feed applications. Ingesting microplastics can damage the gut, causing blockages, abrasion, and cell destruction [144]. However, none of these side effects, like disrupting the gut, causing occlusions, mechanical irritation, or cell loss, were observed in BSFL reared on a diet containing 20% PVC-MPs. In contrast, the larvae exhibited normal development of their gut epithelium and microvilli. In other insect species, MPs may compromise the peritrophic matrix, a semipermeable layer composed of proteins, glycoproteins, and chitin that separates the midgut lumen from the gut epithelium, either through mechanical abrasion or by passing through its porous structure [144]. However, in BSF larvae exposed to PVC-MPs, the peritrophic matrix remained intact and structurally continuous, indicating no disruption. This conclusion is further supported by scanning electron microscopy (SEM) images combined with EDX microanalysis, which confirmed that BSF larvae were capable of ingesting PVC-MPs without compromising their gut structure [13]. Additionally, plasticizers such as DINP are absorbed from the substrate rather than through direct ingestion of plastic particles. The digestion of microplastic particles by BSFL considers several factors that need to be further studied to demonstrate the potential of *Hermetia illucens* larvae as a bioremediation solution for plastic.

While the ingestion of PVC-MPs did not lead to complete degradation of the plastic polymers during the digestive process, certain biotransformation products were detected. The presence of secondary oxidative biotransformation products of DINP, such as hydroxylated MINP (OH-MINP) and carboxylated MINP (cx-MINP), was found in the frass (larval excrement), suggesting a microorganism-mediated degradation process [13]. These findings support the idea that while BSF larvae can reduce the size of ingested microplastic particles, complete degradation of polymers like PVC is beyond their capability [118]. Plastic ingestion has also been linked to the increase in plastic-degrading enzymes within the larvae’s microbiome. Enzymes such as DyP-type peroxidases, multicopper oxidases, and alkane monooxygenases, which are associated with the breakdown of polymer chains, were significantly upregulated in larvae fed on polyethylene (PE) and polystyrene (PS) substrates. This indicates that while the larvae themselves may not fully degrade plastics, their gut microbiota may play a critical role in facilitating polymer breakdown [16]. Additionally, 1H nuclear magnetic resonance (NMR) and ultrastructural analyses confirmed changes in the surface properties of plastic particles, further supporting the hypothesis of microbiome-driven plastic degradation.

#### 5.2.2. Microbiome Alterations Induced by Microplastic Ingestion

The ingestion of microplastics, particularly PVC-MPs, has been shown to impact the composition of the gut microbiota of BSF larvae. While the overall alpha- and beta-diversity of the bacterial and fungal communities did not show significant changes, the presence of PVC-MPs led to a taxon-dependent shift in the relative abundances of certain microbial families [118]. Specifically, the relative abundance of bacteria from the family *Enterobacteriaceae* was significantly higher in larvae reared on a diet containing 20% PVC-MPs, while the family *Paenibacillaceae* was enhanced in both 2.5% and 20% PVC-MP diets. Among fungi, *Dipodascaceae* decreased significantly in response to 20% PVC-MPs, while *Plectosphaerellaceae* exhibited a fluctuating response, decreasing at lower PVC concentrations and increasing at higher levels [118]. The ingestion of microplastics also triggered physiological responses in the larvae, including gut epithelial damage and inflammation, accompanied by the activation of antioxidant enzyme systems to counteract reactive oxygen species (ROS) generated during gut peristalsis and friction with microplastics [145]. In addition to these inflammatory responses, pathogenic bacteria, such as members of the families *Enterococcaceae* and *Clostridia*, increased in the larvae’s gut, which raises concerns about the potential for antibiotic-resistant gene transfer. These findings suggest that while BSF larvae can process microplastics to some extent, the microbial and physiological consequences of such ingestion must be carefully considered. Some studies suggest that the microbiome may help with certain metabolic activities, but its effect on breaking down polymers is limited [17]. Shotgun metagenomics of Black Soldier Fly larvae revealed that plastic feeding not only reshaped the microbiome at the species level (e.g., enrichment of *Gordonia* and *Sphingobacterium*) but also increased genes encoding DyP-type peroxidases. In particular, the phyla *Actinobacteria* and *Verrucomicrobia* became dominant in larvae reared on polyethylene (PE) and polystyrene (PS) substrates [16]. The presence of specific plastic-degrading bacteria, such as *Microbacterium* spp. and *Sphingobacterium* sp., suggests a microbiome-driven adaptation to plastic-rich diets. Moreover, genes encoding key enzymes involved in polymer degradation, such as alkane monooxygenases and multicopper oxidases, were significantly enriched in larvae exposed to plastic substrates, further indicating the functional capacity of the microbiome to degrade complex polymer structures [16]. In the same manner, a study that used BSFL as bio-incubators identified seven isolates that possessed genes associated with plastic degradation pathways. Specifically, the study found genes for PHB/PHA (Polyhydroxyalkanoates/Poly (3-hydroxybutyrate) depolymerases in *Lysinibacillus* sp. 4Z, *Methylobacterium* sp. 4A-1, and *Brevundimonas* sp. 5Z, while 3HV (3-Hydroxyvalerate) dehydrogenase genes were present in all four *Ochrobacterium intermedium* genomes. Additionally, they identified genes for other plastic types. The genome of Methylobacterium sp. 4A-1 contained genes for PLA (Polylactic Acid) degradation, and *Lysinibacillus* sp. 4Z had genes related to nylon degradation. The study also discovered genes for PEG (Polyethylene Glycol) degradation in *Brevundimonas* sp. 5Z and *Ochrobacterium* spp. Overall, these findings indicate that all seven isolates possess the genetic potential to break down a variety of different plastics [15].

Identifying and optimizing the specific microbial populations responsible for polymer degradation could open new avenues for bioremediation efforts in plastic-polluted environments. However, the risk of bioaccumulation of microplastic particles and their associated chemical additives within the larvae must be carefully managed, particularly if they are intended for use in animal feed [13]. Microbial community changes that are observed after exposure to plastic do not directly prove that the microbes are metabolically involved in polymer degradation. While these shifts might suggest a selection for potentially relevant species, a definitive link to actual degradation pathways requires omics-based analyses (like metagenomics, transcriptomics, or proteomics) combined with enzyme characterization.

Research on the impact of microplastics on the black soldier fly larvae (BSFL) microbiome is still in its nascent stages. In contrast, studies on other insect species, such as the waxworm (*Galleria mellonella*), are more advanced. Waxworms are known for their ability to break down polyethylene (PE). This degradation process is thought to be mediated by specific enzymes or the microbial community within the larval gut, which work synergistically to depolymerize the tough plastic material [146].

Larvae of the genus *Zophobas*, particularly the superworm (*Zophobas morio/atratus*), have demonstrated a significant capacity to biodegrade microplastic particles. This effect, documented in studies on polypropylene and polystyrene, is primarily attributed to the enzymatic activity of the larvae’s gut microbiome. Research has shown that plastics ingested by these larvae undergo limited depolymerization, preferentially affecting the lowest molecular weight polymer chains, without displaying signs of significant oxidation. Furthermore, this process is accompanied by notable alterations in the composition of the gut microbiota, indicating a selective pressure driven by the plastic diet [147].

#### 5.2.3. Impact of Microplastics on BSFL Growth and Development

Microplstics (MPs) accumulated solely in the larval gut did not affect the growth and development of BSFL. Larvae efficiently excreted MPs before reaching the pupation stage, suggesting their ability to reduce the potential harm caused by MP accumulation [148,149]. This effective excretion of MPs by BSFL before pupation supports their potential safe use as animal feedstock (Figure 4). However, a careful evaluation of the effects of BSFL reared on contaminated substrates containing non-detectable residues like nanoplastics, chemicals, or toxic metals remains crucial [72,145]. The BSFL’s capacity to excrete MPs before pupation, without affecting their development or mortality, highlights their promise for sustainable waste management and livestock farming [150]. While the role of the BSFL gut microbiome in processing MPs is still under exploration, studies show that MPs do not negatively affect growth and performance [148,151]. However, further studies are needed to evaluate the potential impacts of BSFL on the soil and food systems through the use of frass as an organic fertilizer. In controlled experiments, no significant differences were observed in the development, growth rate, pupation, mortality, or morphology of BSFL reared on MP-contaminated substrates compared to control groups [148]. MPs of various polymer types, including polyethylene (PE), polypropylene (PP), polyvinyl chloride (PVC), and polystyrene (PS), were tested, and no significant impact on BSFL growth performance was reported [148,151]. However, in 2020, C. Xu identified alterations in the gut microbiome, specifically with PVC MPs, which negatively affected the digestion of organic matter, highlighting the potential for MPs to disrupt nutrient processing in the larval gut [152]. There were some variations in larval response to MPs. For instance, BSFL reared on PA (polyamide) showed lower weight at certain stages, but pupae exhibited the highest average weight, an atypical finding suggesting that MPs may influence developmental dynamics [145]. Romano & Fischer observed a reduced pupation rate in larvae exposed to MPs, despite no changes in other parameters [148]. Furthermore, Heussler et al. [150] in 2024 proposed that faster development in smaller larvae could indicate stress responses, such as nutrient deficiencies in the substrate, potentially exacerbated by MP contamination. Some studies suggested the positive effects of MPs on BSFL growth. Heussler proposed that MPs might act as bulking agents, improving substrate aeration and reducing clumping, thereby promoting a better growth environment. However, it is widely accepted that BSFLs do not degrade MPs but simply excrete them before pupation [153]. Therefore, careful evaluation of frass produced by BSFL reared on MP-contaminated substrates is crucial before considering it for agricultural applications due to the risk of MPs entering the soil and food chain [150]. Although no detectable degradation of MPs by BSFL has been observed, the larvae exhibit a limited capacity to metabolize certain organic compounds, such as plasticizers [17] suggested that BSFL might partially biotransform plasticizers like diisononyl phthalate (DINP) into metabolites such as monoisononyl phthalate (MINP) through gastrointestinal hydrolysis. This process appears similar to mechanisms observed in higher organisms, such as humans and rodents. Despite these biotransformations, the overall degradation of MPs is minimal, and BSFL gut microbiota plays a limited role in plastic degradation [17]. Interestingly, studies on the effects of different types of MPs in the larval diet, such as PS and PE, revealed contrasting results. For example, BSFL reared on food waste containing 5% PS exhibited a significantly lower survival rate and reduced substrate consumption, whereas PE-contaminated substrates did not affect survival, although pupation rates increased with higher PE concentrations. Further, the combined presence of PE and PS and salinity (NaCl) in substrates adversely affected BSFL growth, indicating that substrate salinity, rather than MPs alone, may inhibit growth [151]. The BSFL’s resistance to MP toxicity, especially in growth, was consistent across various studies. For example, in 2024, Wang et al. [18] demonstrated that despite the presence of harmful plastic additives like DMP and DOP, BSFL growth and development were unaffected. This robustness may be linked to their microbiota-mediated metabolic capacity, which enables them to withstand and degrade toxic substances. However, MPs such as PP and PE are persistent and resistant to natural degradation, posing a long-term environmental challenge [18]. In summary, while BSFL can tolerate and excrete MPs without significant impacts on their growth or development, the presence of MPs in rearing substrates may affect nutrient processing and larval microbiota composition. Furthermore, MPs and associated plasticizers are resistant to biodegradation, and BSFL does not substantially reduce their size or quantity. Thus, the environmental implications of using BSFL for waste management, particularly in the presence of MPs, must be carefully considered. Future research should focus on understanding the role of BSFL gut microbiota in processing MPs, assessing the safety of frass as a soil amendment, and exploring the potential for BSFL to metabolize or degrade plastic additives effectively.

## 6. Current Research and Findings

### 6.1. Effects of Microplastics on Insect Larvae: Retention, Growth, and Survival Across Diverse Environmental Conditions

Several studies have examined the effects of microplastic ingestion on BSF larvae under various environmental conditions, particularly focusing on microplastic retention, excretion, and impacts on growth and survival (Table 1). Notably, two studies conducted under similar conditions (27 °C and 60% relative humidity) analyzed the ingestion of polyamide (PA) and polylactic acid (PLA) microplastics, both with sizes of less than 150 µm. These studies, using histology and fluoroscopic microscopy, found that PA and PLA were retained only in the gut and were excreted before pupation, without affecting larval growth [151].

Three studies examining the biotransformation and retention of plasticizers, including diisononyl phthalate (DINP), di(2-ethylhexyl) terephthalate (DEHT), and polyvinyl chloride (PVC), provided key insights into their metabolic fate. DINP underwent biotransformation into monoisononyl phthalate (MINP), while DEHT showed no clear metabolic conversion. PVC exhibited no specific retention or growth effects [17]. Quantifying studies involving polyethylene (PE) and polystyrene (PS) microplastics reported size ranges of 400 µm and 500 µm, respectively. The results indicated that larvae exposed to PE exhibited lower weight, increased pupation, and reduced consumption, while PS exposure resulted in higher larval weight, lower survival rates, and reduced substrate consumption [151].

Other studies using stable conditions and analyzing polyethylene (PE) and polypropylene (PP) microplastics, ranging from 125 to 150 µm, utilized larval gut DNA extraction techniques. These studies observed prolonged larval stages but minimal impacts on survival. Further research on dimethyl phthalate (DMP) and dioctyl phthalate (DOP) plasticizers also confirmed minimal effects on larval development [18].

Under stable conditions of 27 °C and 60% relative humidity, PE microplastics with a Dv (50) size of 61.5 µm were found to be excreted without accumulation in the gut, with no impact on larval growth [13]. Similarly, PVC microplastics ranging from 150 to 190 µm showed no accumulation in the gut, and no significant changes in survival were observed, but smaller pupae rates were noticed [118].

In contrast, polypropylene (PP) microplastics with a size of 55 ± 4 µm resulted in delayed pupation and increased fatty acid levels [148]. Additionally, in dark conditions with a temperature of (27.0 ± 0.5) °C and (70 ± 5)% relative humidity, larger microplastic particles (400–1000 µm for PE and 400–800 µm for PS) delayed larval development [16].

### 6.2. Challenges and Limitations in Current Research

Studies on BSF larvae and microplastic ingestion have shown mixed results, making it difficult to draw definitive conclusions. While one study [150] suggests that PA and PLA microplastics were excreted before pupation without growth effects, reported that PE and PS microplastics influenced larval growth. In addition, the contrasting findings on the gut health of BSF larvae exposed to PVC microplastics may be directly attributed to the significant difference in particle size. While one study [13] found that PVC particles with a median diameter of 61.5 µm caused no disruption to the peritrophic matrix or gut structure, another study [145] using much smaller PVC particles (200 nm) observed gut epithelial damage, inflammation, and an increase in pathogenic bacteria. This suggests that the nanoscale dimensions of the particles, rather than the polymer type itself, may be a critical factor in triggering a harmful physiological response, potentially due to their ability to more easily penetrate biological barriers and induce a stronger inflammatory reaction. These inconsistencies may appear from variations in environmental conditions, particle size, and microplastic types, complicating the assessment of their impact.

Although BSF larvae can metabolize certain plastic additives, their ability to break down microplastics is still poorly understood. Another study [17] suggests that DINP was converted to monoisononyl phthalate (MINP), but DEHT showed no clear biotransformation. This suggests that while some compounds undergo metabolic conversion, others may persist, raising concerns about the larvae’s long-term efficiency in microplastic degradation. Many current studies only show that additives like plasticizers are transformed, while direct evidence of polymer mass loss or mineralization is absent. This highlights a serious need for standardized methodologies to accurately quantify polymer degradation beyond just additive metabolism.

While BSF larvae excrete microplastics, concerns remain about the accumulation of microplastics and additives in their frass, which could enter the soil and food chain [150,151]. The potential risks associated with using BSF frass as fertilizer, especially in polluted environments, need thorough evaluation to prevent unintentional environmental contamination.

## 7. Conclusions

The interactions between BSFL and microplastics reveal promising applications in waste management, bioremediation, and sustainable resource recovery. Current research highlights BSFL’s ability to ingest and excrete MPs, offering potential pathways for mitigating plastic pollution and enhancing environmental sustainability. BSFLs have demonstrated the ability to ingest and excrete microplastic particles, particularly those smaller than 150 µm, showing only minor impacts such as slightly reduced growth rates, stress-induced acceleration of pupation, and some variation in survival. However, the larvae’s capacity for complete degradation of microplastics remains constrained, emphasizing their role more as biological filters that accumulate and remove MPs from contaminated substrates rather than fully breaking them down. The reviewed studies also underscore the complexity of MP interactions with BSFL, with particle characteristics such as size, shape, and polymer type significantly influencing ingestion and retention. Although plasticizers like DINP can be partially biotransformed by BSFL, the overall degradation of microplastics is minimal. Moreover, shifts in gut microbiota composition due to MP exposure suggest a potential for specific bacterial families to contribute to the biotransformation process, though these mechanisms are not yet fully understood. The observed microbial community shifts in response to plastic exposure do not directly prove metabolic involvement in polymer degradation. While these shifts may indicate selective pressures or enrichment of potentially relevant taxa, definitive linkage to functional degradation pathways requires omics-based analyses (e.g., metagenomics, transcriptomics, and proteomics) coupled with enzyme characterization. Despite these promising findings, several challenges and risks must be addressed before large-scale implementation of BSFL in bioremediation efforts. Inconsistencies across studies regarding MP ingestion dynamics, as well as concerns about the transfer of microplastics and associated chemicals through BSFL byproducts such as frass, present significant ecological and agricultural risks. Furthermore, the potential for MPs and plastic additives to re-enter soil and food systems via frass requires comprehensive evaluation.

Future research should focus on optimizing BSFL’s microbiome to enhance plastic degradation, understanding the long-term environmental impact of BSFL-based systems, and developing safe protocols for their use in agricultural and industrial applications. Synergistic approaches, such as combining BSFL with plastic-degrading microorganisms, offer a promising pathway for improving the efficacy of BSFL in reducing the environmental burden of plastic pollution. As a multi-functional tool for organic waste management, pollution mitigation, and sustainable feed production, BSFL represents a key component in advancing circular economy practices.

## Figures and Tables

**Figure 1 insects-16-00913-f001:**
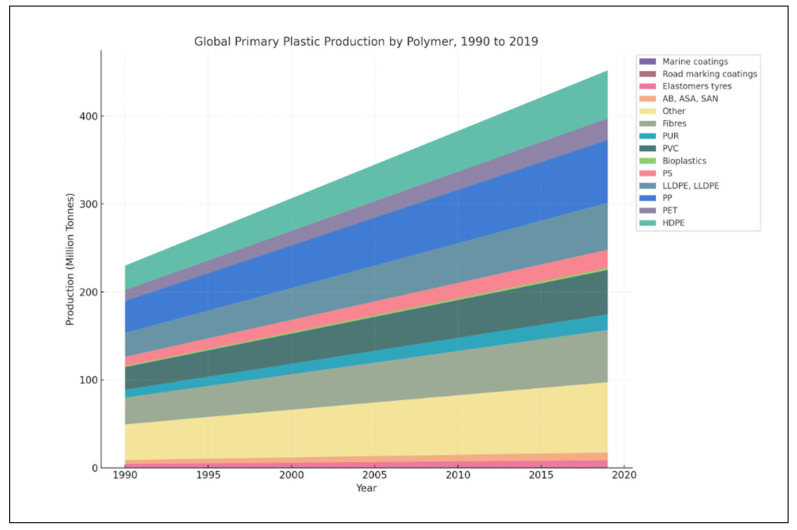
Global Primary Production by Polymer, 1990 to 2019.

**Figure 2 insects-16-00913-f002:**
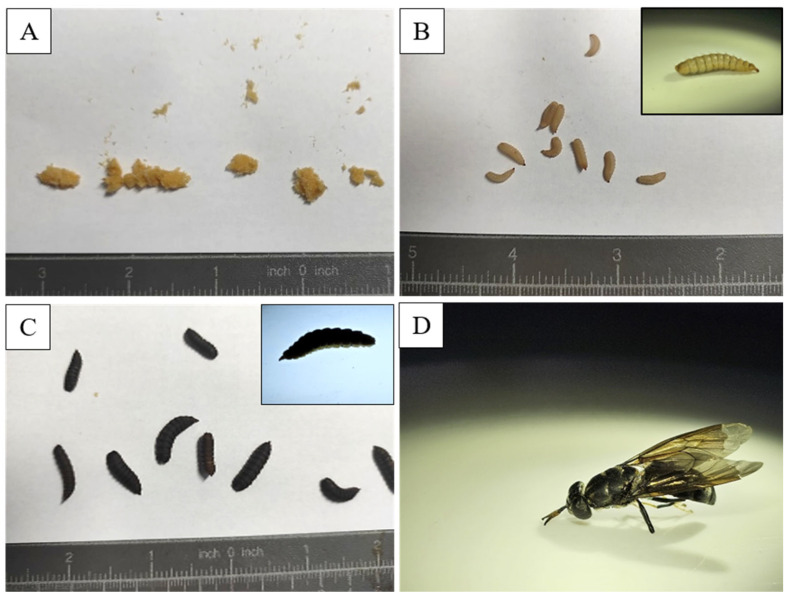
BSFL life cycle. Eggs (**A**), larval stage (**B**), pupae stage (**C**), and adult stage (**D**).

**Figure 3 insects-16-00913-f003:**
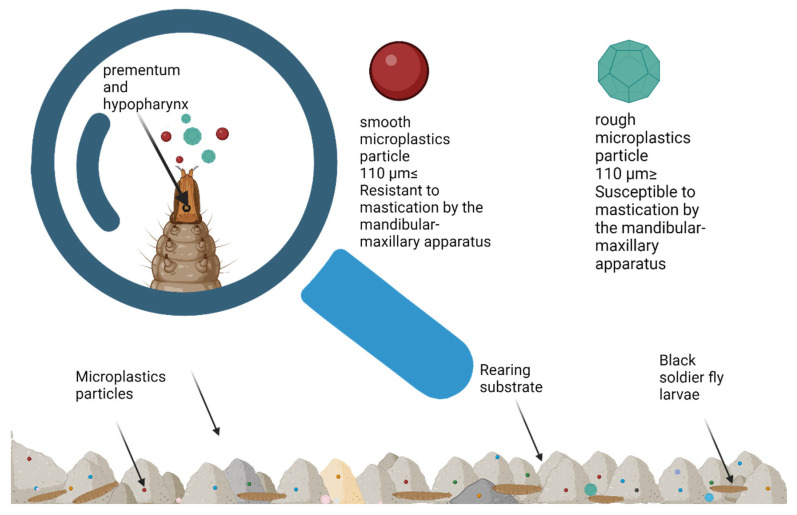
Factors that influence microplastic particles’ physical fragmentation by Black Soldier Fly larvae.

**Figure 4 insects-16-00913-f004:**
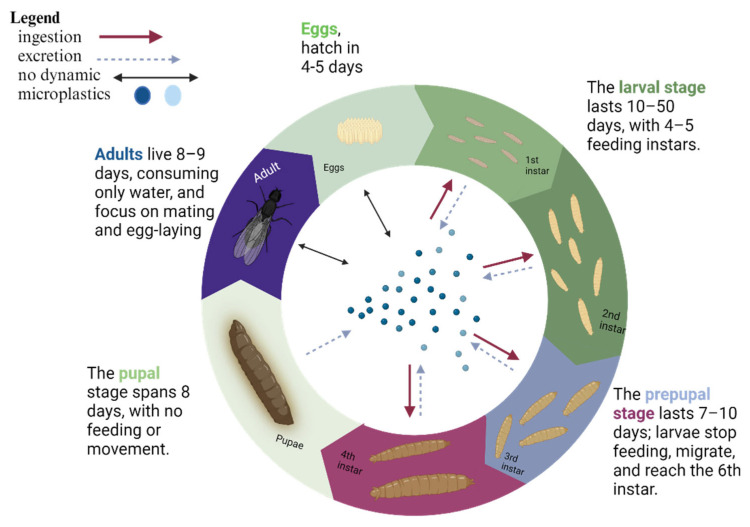
The influence of microplastic particles on a few developmental stages of Black Soldier Fly.

**Table 1 insects-16-00913-t001:** Exploring the Effects of Microplastics on BSF Larvae: Retention, Growth, and Survival Across Diverse Environmental Conditions.

No.	Environmental Conditions	Types of Microplastics Analyzed	Size Range of Microplastics	Analytical Techniques	Microplastic Retention and Excretion	Effects on Growth and Survival	Reference
**1.**	27 °C and 60% relative humidity	PA	(<150 μm)	Histology Fluoroscopic microscopy	MPs only in gut MPs excrete before pupation	Growth unaffected	[150]
**2.**	27 °C and 60% relative humidity	PLA	(<150 μm)	Histology Fluoroscopic microscopy	MPs only in gut MPs excrete before pupation	Growth unaffected	[150]
**3.**	27 °C with a relative humidity of 60%	DINP	Not specified	Gas chromatographic Liquid chromatographic	Moderate DINP intake, converted to MINP	Not specified	[17]
**4.**	27 °C with a relative humidity of 60%	DEHT	Not specified	Gas chromatographic Liquid chromatographic	no clear biotransformation pattern	Not specified	[17]
**5.**	27 °C with a relative humidity of 60%	PVC	Not specified	Gas chromatographic Liquid chromatographic	Not specified	Not specified	[17]
**6.**	Temp: 27–28°C, Humidity: 50%+	PE	400 μm	Quantifying	Not specified	Lower weight on PE Increased pupation, reduced consumption	[151]
**7.**	Temp: 27–28°C, Humidity: 50%+	PS	500 μm	Quantifying	Not specified	Higher weight on PS Lower survival, less substrate	[151]
**8.**	Stable Conditions	PE	125–50 μm	Measurements Larval gut DNA Extraction	Not specified	Larval stage prolonged	[18]
**9.**	Stable Conditions	PP	125–150 μm	Measurements Larval gut DNA Extraction	Not specified	Larval stage prolonged	[18]
**10.**	Stable Conditions	DMP	Not specified	Measurements Larval gut DNA extraction	Not specified	Minimal impact	[18]
**11.**	Stable conditions	DOP	Not specified	Measurements Larval gut DNA extraction	Not specified	Minimal impact	[18]
**12.**	At 27 °C and 60% humidity	PE	(Dv(50) = 61.5 μm)	Fluorescent microscopy SEM	No gut accumulation, particles excreted	Growth unaffected	[13]
**13.**	14-h photoperiod, 28 ± 3 ◦C, relative humidity 60 ± 10 %)	PVC	150–190 μm	SEM, EDX, TEM, qPCR, Bacterial and fungal DNA amplification and metabarcoding sequencing	No gut changes	No mortality rise, smaller pupae	[118]
**14.**	30–40% humidity & 25°C. Temp.	PP	55 ± 4 μm	Measurement SCFA profile gas chromato-Graphy	Not specified	Lower pupation, higher fatty acid levels	[148]
**15.**	27.0 ± 0.5 °C, 70 ± 5% relative humidity, in the dark	PE	400–1000 μm	SEM Metagenomic analyses of the midgut	Not specified	Delayed development	[16]
**16.**	27.0 ± 0.5 °C, 70 ± 5% relative humidity, in the dark	PS	400–800 μm	SEM Metagenomic analyses of the midgut	Not specified	Delayed development	[16]

SEM: Scanning electron microscopy, EDX: energy-dispersive x-ray microanalysis, TEM: Transmission electron microscopy, qPCR-Total: DNA extraction and quantification, MINP: monoisononyl phthalate, MPs: Microplastics, PA: Polyamide, PLA: polylactic acid, DINP: diisononyl phthalate-Plasticizer, DEHT: di(2-ethylhexyl) terephthalate Plasticizer, PVC: polyvinyl chloride, PE: polyethylene, PS: polystyrene, PP: polypropylene, DMP: Dimethyl-phthalate-phthalic acid esters plasticizers, DOP: dioctyl phthalate-phthalic acid esters plasticizers.

## Data Availability

As this is a review article, no new data were generated. The data used is from previously published studies, which are all cited in the references.

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
