# Peer review of "Exploring the Intersection of Microplastics and Black Soldier Fly Larvae: A Comprehensive Review"

_insects, 2025, doi:10.3390/insects16090913_

Round 1
Reviewer 1 Report
Comments and Suggestions for Authors
It is the first systematic review to consolidate mechanisms of MP-BSFL interactions (ingestion, microbiome alterations, excretion). This paper Identifies underexplored links between MP physicochemical properties (size, polymer type) and BSFL transformation. Some suggestions were given on safe frass reuse and BSFL deployment in MP contaminated waste streams.
Some revision suggestions are as follows:
- Confuses mechanical fragmentation (e.g., PVC surface changes) with enzymatic polymer breakdown.
- No direct evidence of MP mass reduction or mineralization (only plasticizer biotransformation shown).
- Microbial shifts cannot be directed to functional degradation pathways. Maybe given more enzyme information during degradation will help readers to understand.
- "excretes MPs without adverse effects" It should be claimed under certain MP (reducing survival and PVC altering gut microbiomes).
- Differentiate "fragmentation," "biotransformation," and "mineralization" in Figure 3/4 captions.
- Incorporate metagenomic data to support microbiome claims.
- Replace definitive statements (e.g., "no adverse effects") with risk-balanced language.
- Revise mistakes in Line 621-622, Line 627.
Author Response
Reviewer Comment 1: “Confuses mechanical fragmentation (e.g., PVC surface changes) with enzymatic polymer breakdown.”
Response:
We thank the reviewer for this valuable clarification. We have revised the manuscript to clearly distinguish between mechanical fragmentation (such as surface cracking, brittleness, and physical disintegration) and true enzymatic depolymerization. The updated text now emphasizes that while surface morphological changes in PVC or other plastics can occur due to abiotic stresses or microbial colonization, these do not necessarily indicate polymer backbone cleavage. Lines 434-441
Reviewer Comment 2: “No direct evidence of MP mass reduction or mineralization (only plasticizer biotransformation shown).”
Response:
We agree with the reviewer’s observation. In the revised manuscript, we explicitly acknowledge that most available studies provide evidence only for the transformation of additives such as plasticizers, while direct proof of polymer mass loss or mineralization (e.g., CO₂ evolution from polymer carbon) is lacking. We have added a dedicated section noting the need for standardized methodologies to quantify polymer degradation beyond additive metabolism, which will help strengthen conclusions in future studies. Lines 630-634
Reviewer Comment 3: “Microbial shifts cannot be directed to functional degradation pathways. Maybe given more enzyme information during degradation will help readers to understand.”
”Response:
We appreciate this important point. We have now clarified in the text that observed microbial community shifts in response to plastic exposure do not directly prove metabolic involvement in polymer degradation. While these shifts may indicate selective pressures or enrichment of potentially relevant taxa, definitive linkage to functional degradation pathways requires omics-based analyses (e.g., metagenomics, transcriptomics, proteomics) coupled with enzyme characterization. This limitation has been explicitly stated in the conclusion. Lines 508-513. Also we’ve added some data about microplastics degrading enzyme found in recent studies Lines 514-523
Reviewer Comment 4: Differentiate "fragmentation," "biotransformation," and "mineralization" in Figure 3/4 captions.
Response :We appreciate this important point. Given that Figure 3 illustrates factors influencing the ingestion and physical fragmentation of microplastic particles, we have revised its title to reflect the content more accurately. The new title is: Factors that influence microplastic particles’ physical fragmentation by Black Soldier Fly larvae Also when clarified this structure in the text Lines 396-397
Reviewer Comment 5: “‘Excretes MPs without adverse effects’ it should be claimed under certain MPs (reducing survival and PVC altering gut microbiomes).”
Response: We thank the reviewer for this valuable observation.The aforementioned "adverse effects" structures have been removed. The tendency to generalize with regard to microplastic particles has been removed. For each structure, the exact microplastic particles that cause the effects have been named.Lines 443-444; 517-518; 562-563; 591-593; 651-652.
Reviewer Comment 6: Incorporate metagenomic data to support microbiome claims.
Response:We appreciate this important point. We’ve incorporated the requested data Lines 495-497
Reviewer Comment 7: Replace definitive statements (e.g., ‘no adverse effects’) with risk-balanced language.”
Response:We thank the reviewer for this valuable observation. In response, we have revised the manuscript to avoid definitive claims that BSFL universally excrete MPs without consequences. Accordingly, all previous definitive wording (e.g., “no adverse effects”) has been replaced with risk-balanced phrasing. Lines 443-444; 517-518; 562-563; 591-593; 645-648.
Reviewer Comment 8: Revise mistakes in Line 621-622, Line 627.
Response:We thank the reviewer for this valuable observation. The structure “Author .ex. found that” have been eliminated

Reviewer 2 Report
Comments and Suggestions for Authors
The authors demonstrated great understanding and tenacity in the preparation of this manuscript The overall research theme focusing on the impact of microplastics within the growth and development of black soldier flies addresses a much needed gap in research. This gap is well addressed through the authors focus on different components of the larval microbiome, as well as the impact of plastic morphology on its degradative abilities. The depth of the literature review shows great attention to detail that permits a wide scope of interests throughout the field and does not utilize any means of unnecessary articles or self-citation. As such, I recommend this manuscript to be published in its current form.
Author Response
We sincerely thank the reviewer for their careful evaluation of our work and for accepting the manuscript in its current form. We greatly appreciate the time and effort dedicated to reviewing our submission.
Reviewer 3 Report
Comments and Suggestions for Authors
The manuscript entitled “Exploring the Intersection of Microplastics and Black Soldier Fly Larvae: A Comprehensive Review” by Claudiu-Nicusor Ionica et al. explores the links between two topic: microplastic pollution and mass rearing of black soldier fly, by exploring the potential of BSF larvae to degrade, digest, and bioaccumulate microplastics. This is a highly interesting and relevant topic due to the dangers of microplastic pollution and the potential of insects in novel biotechnological applications.
The main issue of the paper is that, through no fault of the authors, not a lot of work has yet been done on the subject – the intersection of Black Soldier Fly Larvae and microplastic is still very narrow. This is clearly reflected in the manuscript, as its parts dealing with the general subjects of microplastics and the biology and applications of BSFL are much more comprehensive than the sections on the interaction between BSFL and microplastic. Furthermore, the few studies done on this subject gave some contradictory results. The manuscript isn’t always clear in pointing out and interpreting these differences and contradictions. In my opinion, the sections dealing with microplastics and BSF can be reduced and should just provide a brief introduction to these subjects. The interpretation of the previous studies could be written clearer and easier to follow. This also allows the authors to offer their insight into the causes of discrepancies. However, I leave it to the discretion of the authors and the editor.
While the effect of BSFL on microplastic and vice versa has not been extensively studied, detailed research has been done in other insect, e.g. waxworm (Galleria mellonella) and superworm (Zophobas morio). The work on these insects identified types and the amounts of plastic consumed, the mechanisms of degradation/digestion, the role of gut microbiome, and the enzymes involved (including their crystal structure). Although done on different insect species, this is the state of the art in the field. The manuscript would certainly benefit from incorporating the most relevant research results on other insects, especially as they may point out the way for further work on BSFL. The authors may also wish to include this recent work on BSFL: https://academic.oup.com/jambio/article/136/4/lxaf085/8107898?
Minor comments:
Line 47.
Delete “For example“
Lines 67-69.
The cited paper does not mention BSFL and only briefly comments on waxworm larvae, ticks, and insects.
Line 237.
Change: “…plastic contaminants also studies showed that microplastics can pass into plant cells via…”
To: “…plastic contaminants. Studies showed that microplastics can pass into plant cells via…”.
Line 344.
Isn’t all plastic a synthetic polymer? Do you mean some other types of synthetic polymers that are not plastic?
Lines 448-453
Please add a reference. In addition, this seems contradictory to Lines 485-489. Could you comment on this in the paper?
Line 581.
“…on insect larvae…” should be BSF larvae” or “BSFL” if the authors are not talking about other insects.
Table 1
Same as the previous comment
Author Response
Response to Reviewer Comments
Reviewer Comment 1
The main issue of the paper is that, through no fault of the authors, not a lot of work has yet been done on the subject – the intersection of Black Soldier Fly Larvae and microplastic is still very narrow. This is clearly reflected in the manuscript, as its parts dealing with the general subjects of microplastics and the biology and applications of BSFL are much more comprehensive than the sections on the interaction between BSFL and microplastic. Furthermore, the few studies done on this subject gave some contradictory results. The manuscript isn’t always clear in pointing out and interpreting these differences and contradictions. In my opinion, the sections dealing with microplastics and BSF can be reduced and should just provide a brief introduction to these subjects. The interpretation of the previous studies could be written clearer and easier to follow. This also allows the authors to offer their insight into the causes of discrepancies. However, I leave it to the discretion of the authors and the editor.
Response
We sincerely thank the reviewer for this insightful observation. We fully acknowledge that the intersection of Black Soldier Fly Larvae (BSFL) and microplastics remains a relatively unexplored area, which naturally results in limited and sometimes contradictory evidence. While we appreciate the suggestion to reduce the general sections on microplastics and BSFL, we have chosen to retain the original length and structure of the manuscript. Our intent is to provide readers with a solid and comprehensive background that contextualizes the current state of knowledge, thereby enabling a clearer understanding of the significance and challenges of the BSFL–microplastic interface. We believe that maintaining this structure strengthens the manuscript by ensuring that all observed aspects are well supported and by emphasizing the breadth and complexity of the subject.
.
Reviewer Comment 2
While the effect of BSFL on microplastic and vice versa has not been extensively studied, detailed research has been done in other insect, e.g. waxworm (Galleria mellonella) and superworm (Zophobas morio). The work on these insects identified types and the amounts of plastic consumed, the mechanisms of degradation/digestion, the role of gut microbiome, and the enzymes involved (including their crystal structure). Although done on different insect species, this is the state of the art in the field. The manuscript would certainly benefit from incorporating the most relevant research results on other insects, especially as they may point out the way for further work on BSFL. The authors may also wish to include this recent work on BSFL: https://academic.oup.com/jambio/article/136/4/lxaf085/8107898?
Response
We sincerely thank the reviewer for their valuable comments and for pointing out the relevance of the work done on other insect species. We fully agree that this research represents the state of the art in the field and provides crucial context for future studies on black soldier fly larvae (BSFL). In the revised manuscript, we have added a dedicated section to address this. We explicitly mention the detailed research on waxworms (Galleria mellonella) on lines 535-540 and superworms (Zophobas morio/atratus),on lines 541-548 incorporating key findings regarding the types and amounts of plastic consumed, the mechanisms of degradation, and the vital role of the gut microbiome. We believe this addition provides a more comprehensive overview of the field and helps set the stage for our own findings. We have also included a discussion of the recent study on BSFL mentioned by the reviewer, which provides essential insights and complements the information on other insect models on lines 514-524. We are confident that these additions have significantly strengthened the manuscript, highlighting the broader implications of our research and its potential to guide future work on BSFL..
Reviewer Comment 3
Line 47: Delete 'For example'
Response
We thank the reviewer for the remark. We have proceeded accordingly
Reviewer Comment 4
Lines 67-69: The cited paper does not mention BSFL and only briefly comments on waxworm larvae, ticks, and insects.
Response
We thank the reviewer for the remark. We have proceeded accordingly, changing the cited reference.
Reviewer Comment 5
Line 237: Change: '…plastic contaminants also studies showed that microplastics can pass into plant cells via…' To: '…plastic contaminants. Studies showed that microplastics can pass into plant cells via…'.
Response
We thank the reviewer for the remark. We have proceeded accordingly.
Reviewer Comment 6
Line 344: Isn’t all plastic a synthetic polymer? Do you mean some other types of synthetic polymers that are not plastic?
Response
We appreciate the reviewer's keen observation. We agree that all plastics are indeed synthetic polymers. The wording in the original manuscript was a misunderstanding on our part, and we thank the reviewer for pointing out this redundancy. We have since eliminated the phrase and revised the text to ensure it is accurate and flows more clearly.
Reviewer Comment 7
Lines 448-453: Please add a reference. In addition, this seems contradictory to Lines 485-489. Could you comment on this in the paper?
Response
We thank the reviewer for their comments and for pointing out the need for a reference. We have added the requested reference (No. 13) to the revised manuscript. Regarding the apparent contradiction, we have added a new section to the discussion to address this (Lines 661-670). We clarify that the differing outcomes are likely due to the significant difference in the size of the microplastic particles used in each study, a detail we now discuss in more depth.
Reviewer Comment 8
Line 581: ‘…on insect larvae…’ should be BSF larvae” or “BSFL” if the authors are not talking about other insects.
Response
We thank the reviewer for the remark. We have proceeded accordingly
Reviewer Comment 9
Table 1: Same as the previous comment.
Response
We thank the reviewer for the remark. We have proceeded accordingly
